# Exploit CAM by itself: Complementary Learning System for Weakly Supervised Semantic Segmentation

**Wankou Yang**                                                          *wkyang@seu.edu.cn*
*School of Automation*
*Southeast University*

**Jiren Mai**                                                            *maijiren@seu.edu.cn*
*School of Automation*
*Southeast University*

**Fei Zhang**                                                            *ferenas@sjtu.edu.cn*
*Department of Electronic Engineering*
*Shanghai Jiao Tong University*

**Tongliang Liu**                                                        *tongliang.liu@sydney.edu.au*
*Sydney AI Centre*
*The University of Sydney*

**Bo Han**                                                               *bhanml@comp.hkbu.edu.hk*
*TMLR Group*
*Hong Kong Baptist University*

**Reviewed on OpenReview:** *https://openreview.net/forum?id=KutEe24Yai*

## Abstract

Weakly Supervised Semantic Segmentation (WSSS) with image-level labels has long been suffering from fragmentary object regions led by Class Activation Map (CAM), which is incapable of generating fine-grained masks for semantic segmentation. To guide CAM to find more non-discriminating object patterns, this paper turns to an interesting working mechanism in agent learning named Complementary Learning System (CLS). CLS holds that the neocortex builds a sensation of general knowledge, while the hippocampus specially learns specific details, completing the learned patterns. Motivated by this simple but effective learning pattern, we propose a General-Specific Learning Mechanism (GSLM) to explicitly drive a coarse-grained CAM to a fine-grained pseudo mask. Specifically, GSLM develops a General Learning Module (GLM) and a Specific Learning Module (SLM). The GLM is trained with image-level supervision to extract coarse and general localization representations from CAM. Based on the general knowledge in the GLM, the SLM progressively exploits the specific spatial knowledge from the localization representations, expanding the CAM in an explicit way. To this end, we propose the Seed Reactivation to help SLM reactivate non-discriminating regions by setting a boundary for activation values, which successively identifies more regions of CAM. Without extra refinement processes, our method is able to achieve improvements for CAM of over 20.0% mIoU on PASCAL VOC 2012 and 10.0% mIoU on MS COCO 2014 datasets, representing a new state-of-the-art among existing WSSS methods. The code is publicly available at: https://github.com/tmlr-group/GSLM.

## 1  Introduction

Semantic segmentation (SS) plays an important role in computer vision, which aims to classify each pixel in an image. Due to the success of deep learning and CNNs, SS has witnessed great progress in recent years, giving

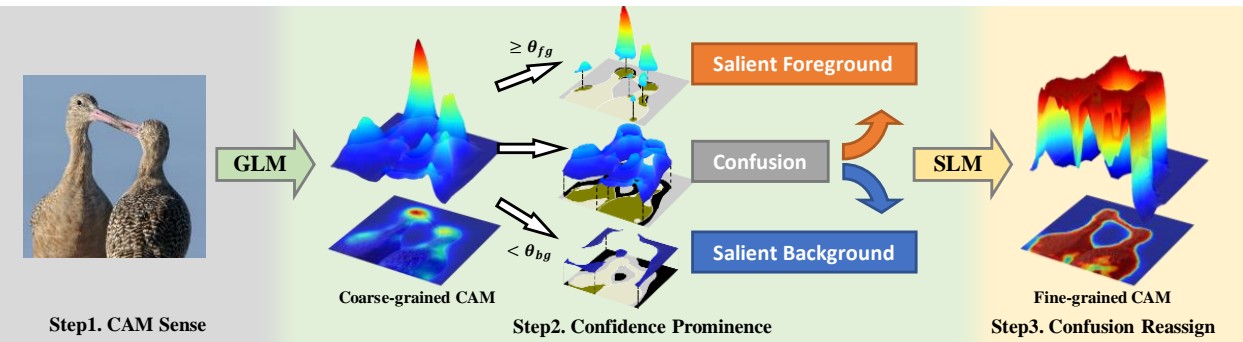

Figure 1: Overview of our method. The General Learning Module (GLM) senses coarse-grained CAM, which is divided into three parts by thresholds of $\theta_{bg}, \theta_{fg}$, and the corresponding local ground-truth is shown below (brown for foreground and black for background). With the guidance of the marked CAM, the Specific Learning Module (SLM) supports confusion reassigning to build a fine-grained CAM.

birth to numerous remarkable works (Chen et al., 2017a;b; Huang et al., 2019; Strudel et al., 2021; Cheng et al., 2021). However, these methods heavily rely on accurate pixel-wise annotations for fully supervised training, which is time-consuming and labor-intensive to collect, making them less economical to apply. Weakly-supervised semantic segmentation (WSSS) is developed to liberate humans from these exhaustive annotation efforts, using weaker and cheaper annotations to achieve semantic segmentation. Image-level labels (Ahn et al., 2019; Zhang et al., 2021; Jiang et al., 2022; Lee et al., 2022), scribbles (Lin et al., 2016), bounding boxes (Dai et al., 2015; Khoreva et al., 2017), and points (Bearman et al., 2016) are some annotation types commonly used for WSSS. Especially, this paper focuses on WSSS based solely on image-level labels.

A key problem in WSSS is how to derive localization cues with only the supervision of image-level labels. By exploiting the contribution of local regions to classification confidences, Class Activation Maps (CAM) (Zhou et al., 2016) provides a key idea and has been commonly used in current WSSS methods (Kolesnikov & Lampert, 2016; Qin et al., 2022; Lee et al., 2021a; Zhang et al., 2021; 2020; Wang et al., 2020). However, CAM is incapable of serving as a fine-grained mask, as it simply captures the small salient regions with discriminating features. To address this issue, many works (Wang et al., 2020; Kim et al., 2021; Zhang et al., 2020; 2021) have tried to drive the CAM to cover more of the target regions. These methods have achieved some success, but the artificially designed modules are complex and it is difficult to integrate their advantages. In this paper, we turn to an interesting concept in agent learning theories, improving CAM based on an experiential learning process.

The Complementary Learning Systems (CLS) (McClelland et al., 1995; Kumaran et al., 2016) hypothesis suggests that two learning systems play complementary roles between the neocortex and the hippocampus in human learning process. The neocortex misses some details but generally builds a perception of patterns in target knowledge, while the hippocampus repeatedly consolidates learned patterns and completes details. We contend that the problem of the non-discriminating regions overlooked by CAM is similar to that of the missing details in the neocortex. As shown by the confidence distribution of the CAM in Figure 1, most pixels in the CAM with neutral confidence are difficult to assign as foreground or background. Indeed, the confusion regions represented by these pixels are mixed with true foreground (brown pixels in the figure) and true background (black pixels). As the salient foreground and background are what CAM has learned, we treat the confusion regions as the missing details of CAM. Therefore, we propose a General-Specific Learning Mechanism (GSLM) to help CAM reactivate the missing details and reassign these confusion regions, similar to the collaborative work of the neocortex and hippocampus in CLS. In GSLM, a General Learning Module (GLM) learns patterns of image-level labels and provides localization representations. A Specific Learning Module (SLM) inherits GLM and adjusts the network weights with these representations, supported by the proposed activation loss (Eq. 5). The representations are termed Confidence CAM and are calculated from the original CAM by our proposed Coarse Generation (Eq. 3), which marks salient and confusion regions with boundary constraints. To achieve the reassignment of the confusion regions, Seed Reactivation (Eq. 4) is integrated into SLM, which uses bounded functions to suppress salient regions and activates non-discriminating regions.

By applying GSLM, a fine-grained CAM can be directly obtained by the conventional CAM network. In addition, we show that GSLM is compatible with other CAM generators (Wang et al., 2020; Lee et al., 2021a; Zhang et al., 2021; Ahn et al., 2019) and can be applied to advanced methods for further improvements.

We set up performance experiments on the PASCAL VOC 2012 (Everingham et al.) and the MS COCO 2014 (Lin et al., 2014) datasets to verify the effectiveness of our method. For accuracy of CAM, our method improves the baseline (48.6% mIoU) by 22.1% mIoU on the PASCAL VOC 2012 *train* set and the baseline (32.5% mIoU) by 10.8$ mIoU on the MS COCO 2014 *train* set, ahead of existing methods over 11.6% mIoU. For the accuracy of pseudo-masks, our method achieves 75.1% mIoU on the PASCAL VOC 2012 *train* set and 43.7% mIoU on the MS COCO 2014 *train* set, which is far beyond existing methods. For segmentation results, our method achieves a new state-of-the-art performance of 70.6% mIoU both on *val* and *test* set of PASCAL VOC 2012 and 40.9% mIoU on the MS COCO 2014 *val* set.

Our main contributions are summarized as follows:

- We propose a simple yet efficient training process, General-Specific Learning Mechanism (GSLM), to drive CAM to reassign the confusion regions, producing fine-grained CAM.

- We apply GSLM to different advanced methods and achieve further improvements on the accuracy of the CAMs.

- Experimental results on both PASCAL VOC 2012 and MS COCO 2014 show that our method outperforms the previous state-of-the-art. In particular, our method boosts baseline CAM by 22.1% mIoU the PASCAL VOC 2012 *train* set, ahead of the best existing methods over 11.6% mIoU.

## 2  Related Work

**Semantic Segmentation.**  Semantic Segmentation aims at assigning a predefined category to every pixel on a given image. After introducing the fully convolutional network (FCN) (Long et al., 2015) into this task, researchers have designed various efficient models to improve performance, including dilated convolutions (Chen et al., 2017a), encoder-decoder architecture (Chen et al., 2017b; Ronneberger et al., 2015), and feature pyramid (Zhao et al., 2017; Chen et al., 2017b). To further promote the model's ability of context aggregation (Zhang et al., 2019; Yuan et al., 2021), self-attention (Huang et al., 2019; Li et al., 2019) paradigms are utilized to learn long-range dependency. Based on this, vision transformers (Strudel et al., 2021; Cheng et al., 2021) are adapted to segmentation tasks, which occupy the state-of-the-art model in different benchmarks. This paper focuses mainly on semantic segmentation in the weakly supervised scenario.

**Weakly Supervised Semantic Segmentation.**  Weakly Supervised Semantic Segmentation (WSSS) aims to generate pixel-level annotation for segmentation tasks. For Image-level based WSSS, most works (Kolesnikov & Lampert, 2016; Qin et al., 2022; Lee et al., 2021a; Zhang et al., 2021; 2020; Wang et al., 2020) have followed a prevailing pipeline to address WSSS, which could be described as 1) training a pseudo-mask generator with image-level labels, and 2) training a fully-supervised semantic segmentation with the generated pseudo-mask labels. Clearly, generating an initial seed map to serve as the pseudo-mask (step 1) is vital for WSSS. To achieve this goal, nearly all works have turned to CAM, which mark out the localization information of an image by merely using a classification network. However, CAM merely focuses on incomplete and fragmentary object regions, driving a gap toward the fine-grained target segmentation maps. To address this issue, there are two mainstream methods to help CAM extract more potential seeds.

The first category mainly focuses on generating better seed regions during the generation of CAM (Wang et al., 2020; Zhang et al., 2020; 2021; Lee et al., 2021a; Chen et al., 2022; Kim et al., 2021; Jiang et al., 2019). Some methods (Wei et al., 2017; Kweon et al., 2021a) focused on erasing the discriminating areas or features in the classification network (Hou et al., 2018; Lee et al., 2019), forcing CAM to pay attention to discriminate other potential object areas. Wei et al. (2017) proposed to deliberately erase specific regions recognized from CAM, and iteratively re-train them for completing object region. Another idea is to artificially add prior knowledge to lead the CAM network to pay attention to potential features. Cross-image mining methods (Fan et al., 2020b; Sun et al., 2020) added cross-image modules that collect semantics on a higher level, from the

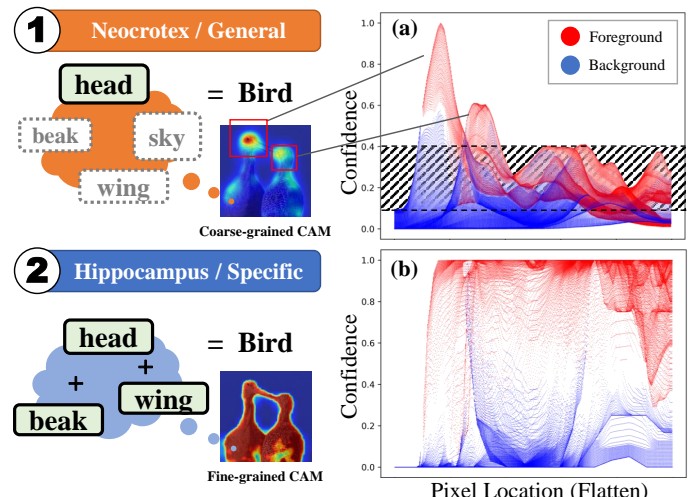

Figure 2: Learning stages of CLS and GSLM. For target patterns of birds, ① neocortex / General Learning Module (GLM) builds a general sense of the target, and ② hippocampus / Specific Learning Module (SLM) further learns details for completing patterns. The distribution of confidence of (a) Coarse-grained CAM and (b) Fine-grained CAM produced from two stages are shown in right.

relation of multiple images with the same classes. Besides, global memory methods (Meng et al., 2021; Jiang et al., 2019) turned to constantly record the category features during the training period, expanding the CAM based on the knowledge learned in previous stages. Furthermore, some works (Wang et al., 2020; Zhang et al., 2021) investigated specific data regularizations to lead the CAM expansion in an explicit way. SEAM (Wang et al., 2020) proposed to leverage the scale invariance of the spatial characteristics while CPN (Zhang et al., 2021) turned to help CAM by using a complementary pair of image inputs.

The second category investigates post-processing to refine the well-trained CAM. Conditional Random Fields (CRF) (Krähenbühl & Koltun, 2011) is one of the most frequently used non-training refiner methods in WSSS. To take full advantage of the pixel-level semantic relationship, PSA (Ahn & Kwak, 2018) and IRN (Ahn et al., 2019) aimed to learn the similarity between pixels obtained from CRF, and apply random walk to further refine the seed areas.

## 3   Preliminaries & Motivation

**Class Activation Map.**   CAM (Zhou et al., 2016) identifies the contribution distribution towards classification predictions, extracted from the classification network, and extensively utilized as the initial localization cues for WSSS methods. Specifically, the classification network for CAM consists of a feature extract backbone, a Global Average Pooling (GAP) and a classification layer. Given the input image $x$ of size $3 \times H \times W$, with a height $H$, width $W$, and $C$ potential categories, we denote the combination of the feature extract backbone and the classification layer as $f : \mathbb{R}^{3 \times H \times W} \to \mathbb{R}^{C \times H \times W}$. The classification prediction score $\hat{y} \in \mathbb{R}^C$ can be calculated as $\hat{y} = \sigma(\text{GAP}(f(x)))$, where $\sigma$ denotes the sigmoid activation function. To train the network, the classification loss is defined as the binary cross-entropy between prediction $y$ and ground-truth $\hat{y}$, i.e.,

$$L_{\text{cls}}(\hat{y}, y) = -\frac{1}{C} \sum_c^C \left( y_c \ln \hat{y}_c + (1 - y_c) \ln (1 - \hat{y}_c) \right). \tag{1}$$

After training $f$, the CAM of an image $x$ related to the $c$-th class, denoted as $M_c \in \mathbb{R}^{C \times H \times W}, c \in \{1, 2, .., C\}$, can be directly calculated as,

$$M_c = \frac{ReLU(f(x)_c))}{\max f(x)_c}. \tag{2}$$

**Shortage of CAM.** In principle, CAM roughly indicates the contribution of local image regions to classification confidence (Ahn et al., 2019). The discriminating features always play a dominant role in the classification, as can be seen from Figure 2(a), where the confidence of the bird head regions (highlighted by red boxes) is much ahead of other regions. The phenomenon results in CAM mainly confined to the small salient regions. Most methods (Wang et al., 2020; Kim et al., 2021; Zhang et al., 2020; 2021) modify the structure of the CAM generating network or introduce well-designed modules, which alter the distribution of image classification confidence to expand the CAM. However, Figure 2(a) suggests that foreground points (in red) generally lie above background points (in blue), indicating that CAM has the potential to distinguish non-discriminating features from the background. Here, we pose the question *Is CAM fully exploited?*

**Fully exploit CAM.** We hold that CAM is not fully exploited by simply training a classification network. As shown by the shaded domain in Figure 2(a), most pixels with neutral confidence in CAM are difficult to accurately assign to either foreground or background, representing "confusion regions". For a classification network, such "confusion regions" can neither symbolize the target nor do not belong to the target. Intuitively, CAM could be enlarged if the pixels of "confusing regions" could be reassigned to the right foreground/background areas, as shown by Figure 2(b). To drive CAM from coarse-grained to fine-grained, we turn to the complementary learning system, one of the famous frameworks in agent learning theory, to help CAM reassign these confusing pixels.

**Complementary Learning System.** We consider the perspective of how our brain learns to address WSSS. Figure 2 shows this process according to CLS (McClelland et al., 1995; Kumaran et al., 2016) theory. Since bird heads appear in most bird images, our neocortex associates the shape of their heads with birds and pays special attention to heads in images. The association is not entirely exact but effective. After several practical trials, the brain expertly picks out the birds head and excludes the sky. In this process, the hippocampus completes more details of the head, causing attention to spread around. In this way, we learn to distinguish birds from a collection of bird images without pixel-level labels. By analogy with CLS, CAM can be considered to be inadequately trained as in the stage of Figure 2①. Our goal is to build a learning mechanism to adequately train CAM. Therefore, we propose the General-Specific Learning Mechanism (GSLM). We develop the General Learning Module (GLM) to act as the neocortex, which generates the coarse-grained CAM and further processes it to serve as a general localization representation of the target. The Specific Learning Module (SLM) is constructed to complete the pattern complement process similar to the hippocampus, which receives guidance from the representations and reactivates non-discriminating features. In recent years, the inspiration of CLS theory on Deep Neural Networks benefits many fields of Artificial Intelligence (e.g. Deep Q-Network (Blakeman & Mareschal, 2020)). It makes a lot of sense to introduce CLS into WSSS.

## 4 Methodology

This section introduces the proposed General-Specific Learning Mechanism (GSLM). In Section 4.1, we illustrate the overall framework of GSLM. Section 4.2 introduces the General Learning Module (GLM), which extracts Confidence CAM with Coarse Generation. Section 4.3 introduces the Specific Learning Module (SLM), into which Seed Reactivation and activation loss are integrated to support pixel-level training and confusion reassignment.

### 4.1 Overall framework of GSLM

GSLM represents a novel training process for a CAM generating network, which consists of a General Learning stage and multiple Specific Learning stages. At each stage, the CAM generating network is wrapped as GLM or SLM, which does not change the topology of the network but affects gradient propagation. Figure 3 shows the GSLM process and the structure of GLM and SLM. Firstly, the network at Stage 1 is wrapped as GLM, which is trained with image-level labels for producing coarse-grained CAM. The coarse-grained will later be refined into Confidence CAM. Secondly, the network at stage 2 is wrapped as SLM, with weights shared from GLM and readjusted under the extra pixel-level supervision of Confidence CAM. The Specific Learning stage

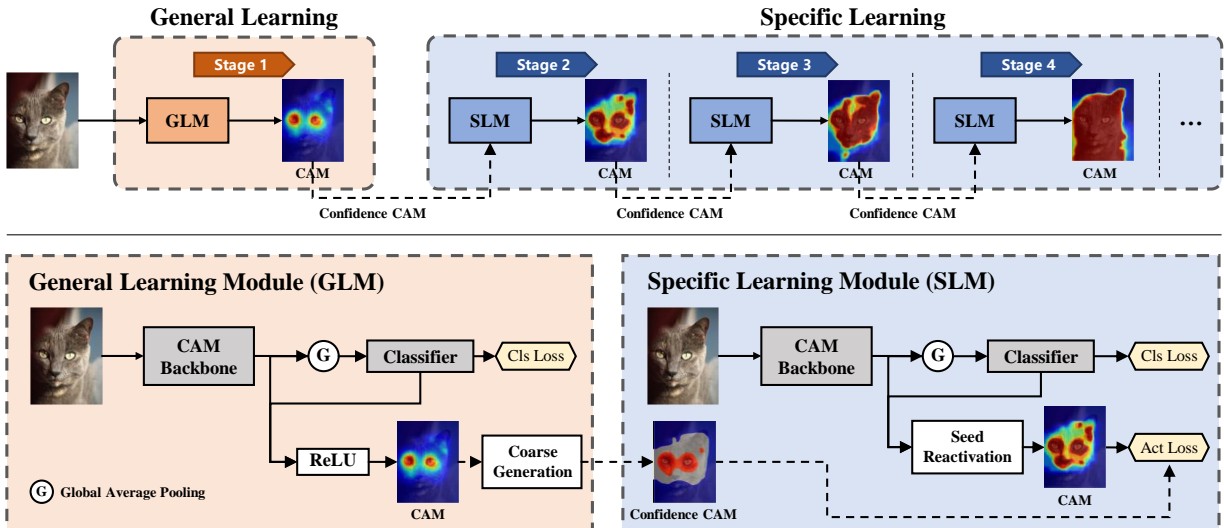

Figure 3: The GSLM framework. In General Learning, the GLM is trained using image-level labels to generate Confidence CAM. Moving on Specific Learning, the SLM is trained using both the image-level labels and the Confidence CAM generated in the previous process. The CAM is refined through GLM multiple stages of SLM, resulting in an improved CAM.

is repeated several times, and specifically, the provider of Confidence CAM and shared weights will be the previous SLM instead of GLM.

As shown by Figure 3, the CAM generation network consists of two branches: classification loss calculation and CAM generation. GLM retains the structure of the original CAM generating network and introduces the Coarse Generation for Confidence CAM generation. SLM uses Seed Reactivation to take over the CAM generation branch of the network and supports activation loss calculation.

## 4.2 General Learning Module (GLM)

As the first stage, GLM aims to generate localization knowledge representation as provision of pixel-level supervision for subsequent stages. Despite carrying localization information, CAM is not suitable for direct use as supervision due to its incomplete object information (Kim et al., 2021; Zhang et al., 2021). We focus on the trusted local of CAM and propose the Coarse Generation algorithm to generate Confidence CAM, an improved localization map that reinforces salient regions and remarks confusion regions, as supervision.

**Confidence CAM.** A Confidence CAM indicates three areas with three values, where salient foreground is set to 1, salient background is set to 0, and rest confusion area is set to -1 as the flag to be ignored. The visualization of a sample Confidence CAM is shown as input of SLM in Figure 3.

**Coarse Generation.** For generating Confidence CAM, we introduce thresholds $\theta_{fg}$ and $\theta_{bg}$ to divide CAM into salient foreground ($\geq \theta_{fg}$), salient background ($< \theta_{bg}$), and confusion area. In addition, conditional random fields (CRF) (Krähenbühl & Koltun, 2011), a widely used non-learnable CAM refinement method, is applied to further refine the salient regions, introducing the boundary constraint information for such knowledge representations. We define a confidence mapping as $g$ and CRF as $\mathcal{R} : \mathbb{R}^{C \times H \times W} \to \mathbb{R}^{C \times H \times W}$. For the $c$-th class, Confidence CAM $N_c$ can be generated from given CAM $M_c$ as follows.

$$N_c = \mathcal{R}(g(M_c)), \quad g(x) = \begin{cases} 1 & , x \geq \theta_{fg} \\ 0 & , x < \theta_{bg} \\ -1 & , \text{else} \end{cases} . \tag{3}$$

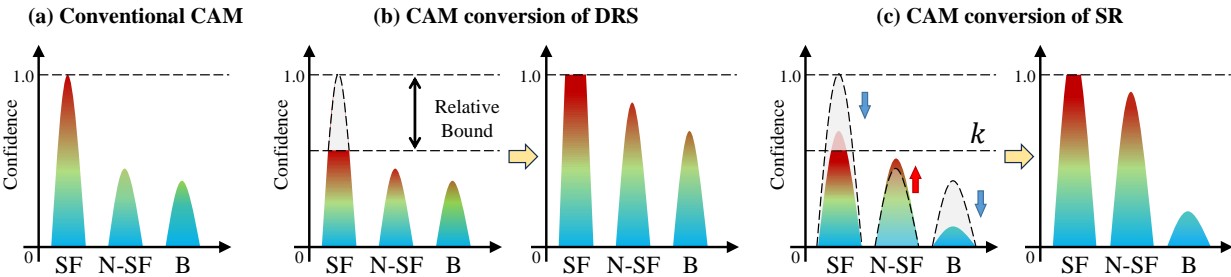

Figure 4: Comparison between DRS (Kim et al., 2021) and Seed Reactivation (SR) of GSLM on refinement principles of CAM. SF: Salient Foreground, N-SF: Non-Salient Foreground, B: Background.

### 4.3 Specific Learning Module (SLM)

SLM aims to readjust the connection weights of CAM generating network with Confidence CAM for Specific Learning. Specifically, Specific Learning refers to consolidating the object patterns of salient regions learned and discovering new patterns to complement the attention of CAM. Therefore, we introduce activation loss, which provides pixel-level supervision with the guiding of Confidence CAM. And we propose Seed Reactivation to help the network activate more regions, free from the crushing effects of salient regions. Besides, the classification loss of GLM is retained for consolidating the learned patterns.

**Seed Reactivation** On the one hand, Seed Reactivation is designed to address the issue that the conventional CAM generation is not compatible with pixel-level supervision based on Confidence CAM. Specifically, the conventional CAM is generated by global maximum normalization (Eq. 2), where all pixels are divided by the maximum pixel value. However, there are cases where the maximum pixel is located in the confusion regions of Confidence CAM and thus ignored during training, failing model convergence:

- SLM may use different weights (and even different network structure) than GLM, and thus the maximum pixel can occur anywhere in CAM.

- Even though SLM and GLM are identical at the beginning of training, and the maximum pixel is plotted in the salient foreground regions of Confidence CAM. However, as the training progresses, SLM will activate some confusion regions as the foreground as expected, and the maximum pixel may be located here.

To address the issue, we introduce a bounded ReLU-k function $\psi$, using a constant denominator for normalization, i.e.,

$$M'_c = \frac{\psi(f(x)_c)}{k}, \quad \psi(x) = \begin{cases} k & , x \geq k \\ 0 & , x < 0 \\ x & , \text{else} \end{cases}. \tag{4}$$

On the other hand, Seed Reactivation (Eq. 4) replaces the conventional CAM generation method (Eq. 2), thereby affecting the resulting CAM. As shown in Figure 4 (a)(c), Seed Reactivation sets a constant upper bound $k$ for regions in the feature map. With pixel-level training, each pixel will be activated (approaching $k$) or suppressed (approaching 0) independently of the other pixels. Similarly, DRS (Kim et al., 2021) applies a relative upper bound to network feature maps to suppress the salient foreground with high confidence. However, the truncation of the high-confidence pixels increases the relative confidence of all remaining pixels, thus incorrectly activating the background region, as shown in Figure 4 (b). For example, a bound of 0.5 relatively increases the confidence of all regions by a factor of 2. The background with high confidence will be misclassified as foreground, resulting in over-activation. By introducing extra supervision saliency maps as background masks, DRS sidesteps the issue of over-activation, but with Seed Reactivation, GSLM can achieve this effect without saliency maps.

**Activation Loss** The activation loss narrows the gap between the improved CAM $M'$ produced by SLM and Confidence CAM $N$ produced by GLM (or SLM in the previous stage). Specifically, we ignore the non-discriminating regions (-1 in Confidence CAM). By measuring the gap with the smooth L1 norm, the activation loss is defined as,

$$L_{\text{act}} = ||M'_i - \boldsymbol{N}_i||_1, \ i \in \{x | \boldsymbol{N}_x \geq 0, x \in \mathbb{R}^{H \times W}\}. \tag{5}$$

Activation loss improves the CAM prediction $M'$ with Confidence CAM $N$ as the target. The foreground confidence will approach 1, while the background confidence will approach to 0, and the confusion area shrinks gradually. The total loss function of SLM is then defined as follows,

$$L = L_{\text{cls}} + \alpha L_{\text{act}}, \tag{6}$$

where $\alpha$ denotes the balance factor between classification loss and activation loss.

## 5 Experiments

### 5.1 Experimental Setting

**Dataset and Evaluation Metrics.** We performed experiments on the PASCAL VOC 2012 dataset (Everingham et al.), a visual object class challenge with 20 categories built for real scenes. It contains a total of 10,582 training images, of which 1,464 images have pixel-level labels and the rest are bounding box labels, 1,449 validation images with pixel-level labels, and 1,456 test images. We also verify our method using the MS COCO 2014 dataset (Lin et al., 2014) which contains 82,783 train and 40,504 validation images with 81 categories. For each dataset, we train our network with the training images and image-level classification labels only, and evaluate the pseudo-masks with pixel-level labels. All image-level labels for training are extracted from pixel-level labels or bounding box labels. The fully supervised semantic segmentation network is trained with the pseudo-masks and evaluated on PASCAL VOC 2012 validation and test set, and MS COCO validation set, respectively. Performance is evaluated by the mean intersection-over-union (mIoU). The evaluation on PASCAL VOC 2012 *test*, without any annotation, is obtained from the official PASCAL evaluation server.

**Implementation details.** We use an ImageNet (Deng et al., 2009) pretrained ResNet50 (He et al., 2016) as the CAM backbone in GSLM and a $1 \times 1$ convolution as the classification layer. For hyper-parameters, $k$ in Eq. 4 is 6, $\alpha$ in Eq. 6 is 0.5, and the thresholds $\theta_{fg}$ and $\theta_{bg}$ in Coarse Generation are 0.30 and 0.05, respectively. The number of iterations of SLM is set to 3 for GSLM. For training setting, the batch size is 16, the training epochs is 5, and the weight decay is 0.001. The remaining optimization settings are the same as in IRN (Ahn et al., 2019). Moreover, we build a stronger baseline GSLM$^+$ by replacing GLM in GSLM with IRN and keeping SLM the same but with a single iteration. All other settings are the same as GSLM.

### 5.2 State-of-the-arts Comparison

**Improvements on CAM.** Table 1 reports the performance comparison of CAM seed and pseudo-masks produced by ours and existing WSSS methods. By convention, pseudo-masks are refined from CAM seeds using PSA (Ahn & Kwak, 2018) or IRN (Ahn et al., 2019). Without refinement, our GSLM reports an excellent result of 67.5% mIoU on the PASCAL VOC 2012 *train* set, outperforming other methods. With the refinement of CRF and IRN (shown as "Mask" in Table 1), GSLM also leads the way and generates pseudo-masks with mIoU of 69.2% and 72.4%, respectively. Figure 5 shows some samples of CAM produced by GSLM. Moreover, we set up GSLM$^+$ by replacing GLM to IRN (reporting 66.3% mIoU in Table 1), achieving a greater performance of 70.7% mIoU (for CAM seed), 73.4% mIoU (for refinement with CRF) and 75.1% mIoU (for pseudo-masks), respectively. Compared with the baseline method IRN, GSLM+ provides an improvement of 8.8% mIoU and is thus the first method to produce CAM seeds with an accuracy exceeding 70% mIoU. Furthermore, GSLM also performs well on the challenging MS COCO 2014 *train* set, reporting 43.3% mIoU for CAM seed and 45.0% mIoU for pseudo-masks.

Table 1: Comparison of pseudo-masks on PASCAL VOC 2012 *train* set and MS COCO 2014 *train* set in mIoU (%). Seed: accuracy of CAM. CRF: accuracy of CAM with CRF. Mask: accuracy of CAM with PSA/IRN. †denotes the re-implemented results.

| Method | VOC2012 | | | COCO | |
|---|---|---|---|---|---|
| | Seed | CRF | Mask | Seed | Mask |
| PSA (Ahn & Kwak, 2018) | 48.0 | - | 61.0 | - | - |
| IRN (Ahn et al., 2019) | 48.8 | 54.3 | 66.3 | 32.5[†] | 38.4[†] |
| CONTA (Zhang et al., 2020) | 48.8 | - | 67.9 | 28.7 | 35.2 |
| SEAM (Wang et al., 2020) | 55.4 | 56.8 | 63.6 | 25.1 | 31.5 |
| CPN (Zhang et al., 2021) | 57.4 | - | 67.8 | - | - |
| AdvCAM (Lee et al., 2021a) | 55.6 | 62.1 | 69.9 | - | - |
| AMR (Qin et al., 2022) | 56.8 | - | 69.7 | - | - |
| CLIMS (Xie et al., 2022) | 56.6 | - | 70.5 | - | - |
| SIPE (Chen et al., 2022) | 58.6 | 64.7 | - | - | - |
| W-OoD (Lee et al., 2022) | 59.1 | 65.5 | 72.1 | - | - |
| PPC (Du et al., 2022) | 61.5 | 64.0 | 70.1 | - | - |
| GSLM (Ours) | 67.5 | 69.2 | 72.4 | **43.3** | **45.0** |
| GSLM[+] (Ours) | **70.7** | **73.4** | **75.1** | 41.4 | 43.7 |

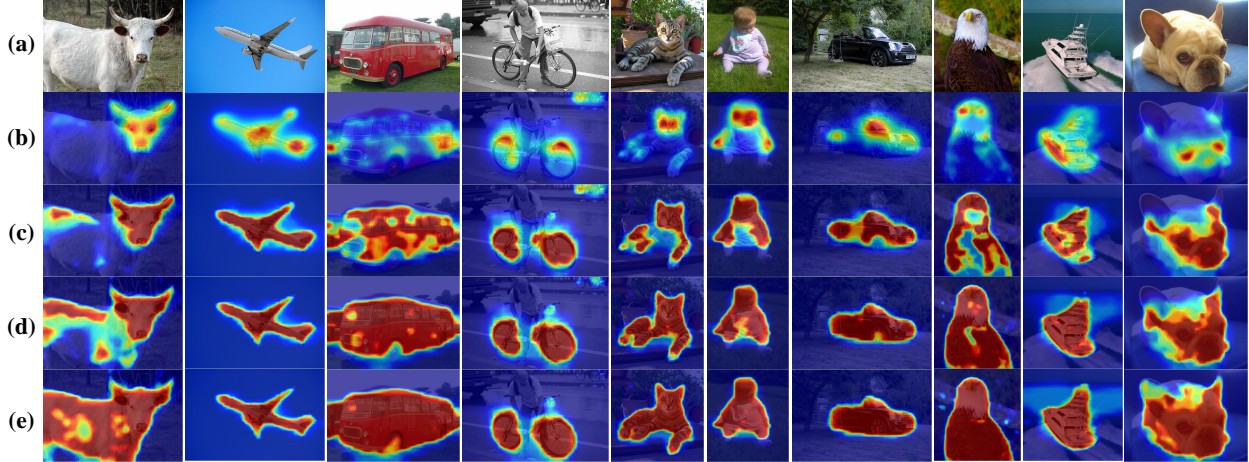

Figure 5: Visualizations of CAM produced by GSLM at different stages on the PASCAL VOC 2012 *train*. (a) Original image. (b) CAM in Stage 1 (equivalent to baseline). (c) CAM after Stage 2. (d) CAM after Stage 3. (e) CAM after Stage 4. With the progress of iterations, GSLM activates broader and more accurate regions.

**Improvements on segmentation results.** To apply our GSLM in semantic segmentation, we train DeepLabV2-ResNet (Chen et al., 2017a) with the pseudo-masks generated by GSLM and GSLM[+], respectively. Table 2 shows the comparison of the performance of state-of-the-art WSSS methods on the PASCAL VOC 2012. Our GSLM achieves 69.4% mIoU on *val* and 69.7% mIoU on *test*. With the seeds provided by IRN, GSLM[+] achieves 70.6% on *val* and 70.6% on *test*, an improvement of 7.1% and 5.8% over IRN. GSLM[+] outperforms existing methods with only image-level labels and most methods with extra supervision, representing new state-of-the-art performance on the PASCAL VOC 2012. GSLM+ is only slightly behind several methods with extra saliency maps (L2G (Jiang et al., 2022) by 1.5% and PPC (Du et al., 2022) 2.0%), validating its excellent performance in constraining over-activation of initial seeds without saliency maps. Figure 6 shows some qualitative samples of semantic segmentation results of our GSLM[+]. Table 3 shows that our GSLM is also efficient for challenging MS COCO 2014 and achieves 40.9% mIoU, ahead of existing methods. Besides, OCR (Cheng et al., 2023) also reports a baseline with MCTformer (Xu et al., 2022) which achieves 72.7% and 72.0% mIoU on PASCAL VOC 2012 and 42.5% mIoU on MS COCO 2014, slightly ahead of GSLM. However, by using DeiT-S (Touvron et al., 2021) as the classification network, OCR-MCTformer

Table 2: Comparison of state-of-the-art WSSS methods on PASCAL VOC 2012 in mIoU (%). Marks of supervision (Sup.) denote image-level labels ($\mathcal{I}$), saliency maps ($\mathcal{S}$) and texts ($\mathcal{T}$). All results are based on ResNet backbone.

| Method | Pub. | Sup. | Val | Test |
|---|---|---|---|---|
| FickleNet (Lee et al., 2019) | CVPR19 | $\mathcal{I}+\mathcal{S}$ | 64.9 | 65.3 |
| ICD (Fan et al., 2020a) | CVPR20 | $\mathcal{I}+\mathcal{S}$ | 67.8 | 68.0 |
| AuxSegNet (Lee et al., 2021b) | ICCV21 | $\mathcal{I}+\mathcal{S}$ | 69.0 | 68.6 |
| NSROM (Yao et al., 2021) | CVPR21 | $\mathcal{I}+\mathcal{S}$ | 70.4 | 70.2 |
| CLIMS (Xie et al., 2022) | CVPR22 | $\mathcal{I}+\mathcal{T}$ | 70.4 | 70.0 |
| L2G (Jiang et al., 2022) | CVPR22 | $\mathcal{I}+\mathcal{S}$ | 72.1 | 71.7 |
| PPC (Du et al., 2022) | CVPR22 | $\mathcal{I}+\mathcal{S}$ | **72.6** | **73.6** |
| IRN (Ahn et al., 2019) | CVPR19 | $\mathcal{I}$ | 63.5 | 64.8 |
| SEAM (Wang et al., 2020) | CVPR20 | $\mathcal{I}$ | 64.5 | 65.7 |
| CONTA (Zhang et al., 2020) | NIPS20 | $\mathcal{I}$ | 66.1 | 66.7 |
| AdvCAM (Lee et al., 2021a) | CVPR21 | $\mathcal{I}$ | 68.1 | 68.0 |
| CPN (Zhang et al., 2021) | ICCV21 | $\mathcal{I}$ | 67.8 | 68.5 |
| PPC (Du et al., 2022) | CVPR22 | $\mathcal{I}$ | 67.7 | 67.4 |
| SIPE (Chen et al., 2022) | CVPR22 | $\mathcal{I}$ | 68.8 | 69.7 |
| W-OoD (Lee et al., 2022) | CVPR22 | $\mathcal{I}$ | 69.8 | 69.9 |
| Kho *et al.* (Kho et al., 2022) | PR22 | $\mathcal{I}$ | 69.5 | 70.5 |
| OCR (Cheng et al., 2023) | CVPR23 | $\mathcal{I}$ | 67.8 | 68.4 |
| GSLM (Ours) | | $\mathcal{I}$ | 69.4 | 69.7 |
| GSLM$^+$ (Ours) | | $\mathcal{I}$ | **70.6** | **70.6** |

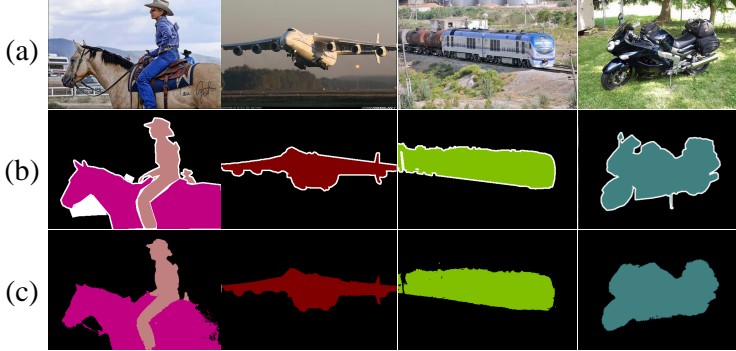

Figure 6: Qualitative results on PASCAL VOC 2012 *val* set. (a) Input Images. (b) Ground-truths. (c) results of GSLM$^+$ (w/ CRF).

cannot be compared fairly with the methods in Table 2 and Table 3 that use ResNet, so it is not included. BECO (Rong et al., 2023) works on robust learning for pseudo-label on WSSS and performs well on PASCAL VOC 2012 (72.1% and 71.8% mIoU) and MS COCO 2012 (45.1%), but it is based on a modified (co-training) DeepLabV3+ (Chen et al., 2018) as a fully supervised segmentation network and is therefore not listed. BECO has made a lot of efforts in training and inference of fully supervised segmentation networks, which can be applied to boost the listed methods, but cannot be fairly compared with them using common DeepLab.

### 5.3 Ablation Study

**Effectiveness on different baselines.** Our GSLM generates CAM seeds and then refines them from 48.6% mIoU to 67.5% mIoU, showing excellent performance. Furthermore, GSLM could also be applied to various WSSS frameworks. As shown in Table 4, GSLM greatly improves the accuracy of the CAM seeds for various WSSS methods. Specifically, it boosts SEAM to 67.6% mIoU (+12.2% mIoU), CPN to 66.9% (+9.6%

Table 3: Comparison of state-of-the-art WSSS methods on MS COCO 2014 in mIoU (%). Marks of supervision (Sup.) denote image-level labels ($\mathcal{I}$) and saliency maps ($\mathcal{S}$). Methods marked by * use VGG backbone and marked by † use ResNet backbone.

| Method | Pub. | Sup. | Val |
|---|---|---|---|
| *ADL (Choe et al., 2020) | PAMI20 | $\mathcal{I}+\mathcal{S}$ | 30.8 |
| †AuxSegNex (Lee et al., 2021c) | ICCV21 | $\mathcal{I}+\mathcal{S}$ | 33.9 |
| †EPS (Lee et al., 2021c) | CVPR21 | $\mathcal{I}+\mathcal{S}$ | 35.7 |
| *L2G (Jiang et al., 2022) | CVPR22 | $\mathcal{I}+\mathcal{S}$ | 42.7 |
| *SEC (Kolesnikov & Lampert, 2016) | ECCV16 | $\mathcal{I}$ | 22.4 |
| †IRN (Ahn et al., 2019) | CVPR19 | $\mathcal{I}$ | 32.6 |
| †CSE (Kweon et al., 2021b) | ICCV21 | $\mathcal{I}$ | 36.4 |
| *RCA (Zhou et al., 2022) | CVPR22 | $\mathcal{I}$ | 36.8 |
| †OCR (Cheng et al., 2023) | CVPR23 | $\mathcal{I}$ | 33.2 |
| †GSLM (Ours) | | $\mathcal{I}$ | **40.9** |

Table 4: The refining effectiveness of GSLM with initial seed produced by different baseline in mIoU (%) on PASCAL VOC 2012 *train* set.

| Method | Seed | +GSLM |
|---|---|---|
| CAM (Zhou et al., 2016) | 48.6 | 67.5$_{+18.9}$ |
| SEAM (Wang et al., 2020) | 55.4 | 67.6$_{+12.2}$ |
| CPN (Zhang et al., 2021) | 57.3 | 66.9$_{+9.6}$ |
| AdvCAM (Lee et al., 2021a) | 55.5 | 68.0$_{+12.5}$ |
| IRN (Ahn et al., 2019) | **66.5** | **70.7**$_{+4.2}$ |

mIoU), and AdvCAM to 68.0% (+12.5% mIoU), all significantly ahead of existing state-of-the-art methods in Table 1. In particular, GSLM improves the performance of IRN up to 70% mIoU, providing fairly high quality CAM seeds for WSSS.

**Effect of main modules.** To evaluate the effect of each part of GSLM, we remove one of Coarse Generation (CR), Seed Reactivation (SR), and classification loss ($L_{cls}$) in Eq (1), respectively, and calculate the mIoU of GSLM, as shown in Table 5. When CR or SR is removed, GSLM loses its usefulness and the performance drops to 49.3% mIoU and 49.5% mIoU. When $L_{cls}$ is removed, performance decreased by 1.5% compared to the complete GSLM. In other words, CR and SR are critical modules, and $L_{cls}$ improves the performance of GSLM. Moreover, by applying SLM iteratively, the result further achieves a 7.4% improvement.

**Effect of boundary constraint in coarse generation.** In Section 4.2, we contend that CRF used in Coarse Generation introduces boundary constraint information to Confidence CAM for advantage of reactivation in SLM. To validate this view, Figure 7 shows the visual comparison of GSLM with and without

Table 5: The ablation study for each part of GSLM on PASCAL VOC 2012 *train* set. CR: Coarse Generation. SR: Seed Reactivation. $L_{cls}$: classification loss in Eq (1). Iter: Iteration of SLM.

| baseline | CR | SR | $L_{cls}$ | Iter | mIoU (%) |
|---|---|---|---|---|---|
| ✓ | | | | | 48.6 |
| ✓ | | ✓ | ✓ | | 49.3 |
| ✓ | ✓ | | ✓ | | 49.5 |
| ✓ | ✓ | ✓ | | | 58.6 |
| ✓ | ✓ | ✓ | ✓ | | 60.1 |
| ✓ | ✓ | ✓ | ✓ | ✓ | **67.5** |

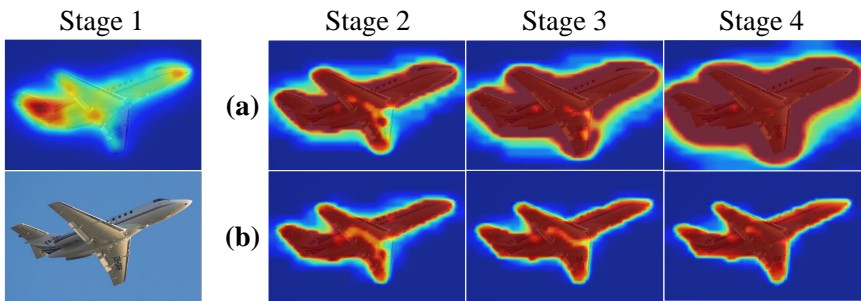

Figure 7: Comparison of CAM changes at different stages of GSLM without (a) and with (b) boundary constraint (BC).

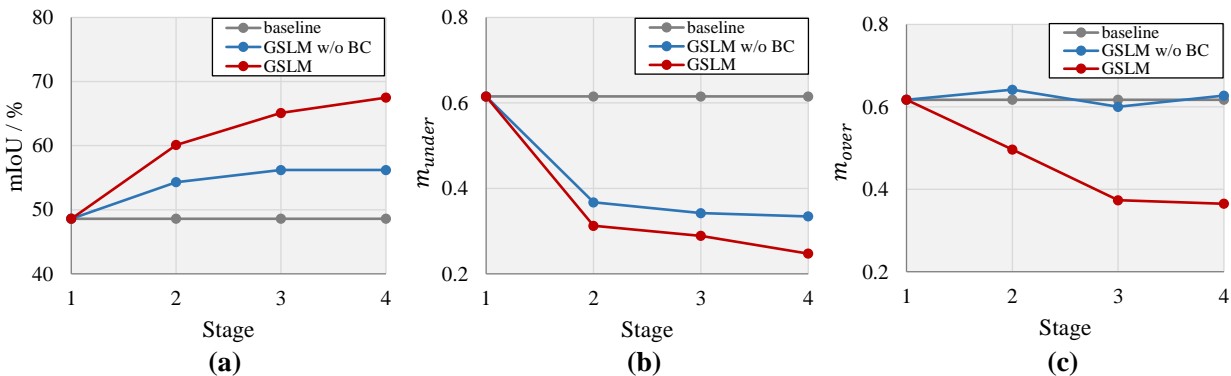

Figure 8: Comparison of GSLM with and without boundary constraint (BC) on (a) CAM accuracy, (b) under-activation, and (c) over-activation in different stages. For (b), the lower the degree of under-activation, the more regions of objects are activated by the network. For (c), the lower the degree of over-activation, the more accurate the network activation area is.

boundary constraint (by adopting CRF in Coarse Generation or not). Figure 7(a) demonstrates that, with the progress of iteration, the activated foreground of GSLM w/o BC is excessively diffused to the background region. This diffusion impedes the enhancement of accuracy, which can be prevented by BC, as shown in Figure 7(b). For further validation, Figure 8 reports the comparison of GSLM w/ BC and GSLM w/o BC in metrics of CAM accuracy, under-activation, and over-activation, respectively. Specifically, we evaluate the degree of under-activation and over-activation by following metrics proposed in (Wang et al., 2020):

$$m_{under} = \frac{1}{C-1} \sum_{c=1}^{C-1} \frac{FN_c}{TP_c} \tag{7}$$

$$m_{over} = \frac{1}{C-1} \sum_{c=1}^{C-1} \frac{FP_c}{TP_c}, \tag{8}$$

where $C$ denotes the number of categories with background, $TP_c$ denotes the pixel number of the true positive prediction of class $c$, $FP_c$ and $FN_c$ denote false positive and false negative, respectively.

In Figure 8(a), GSLM with BC demonstrates better performance than GSLM without BC in all rounds. Figure 8(b) and Figure 8(c) report the changes of the degree of under-activation and over-activation of the two settings of GSLM in the iteration process, respectively. Both of them show similar performance in the optimization of under-activation. However, in the absence of boundary constraints, GSLM fails to control the over-activation degree, as shown in Figure 8(c), the over-activation degree of GSLM without BC is close

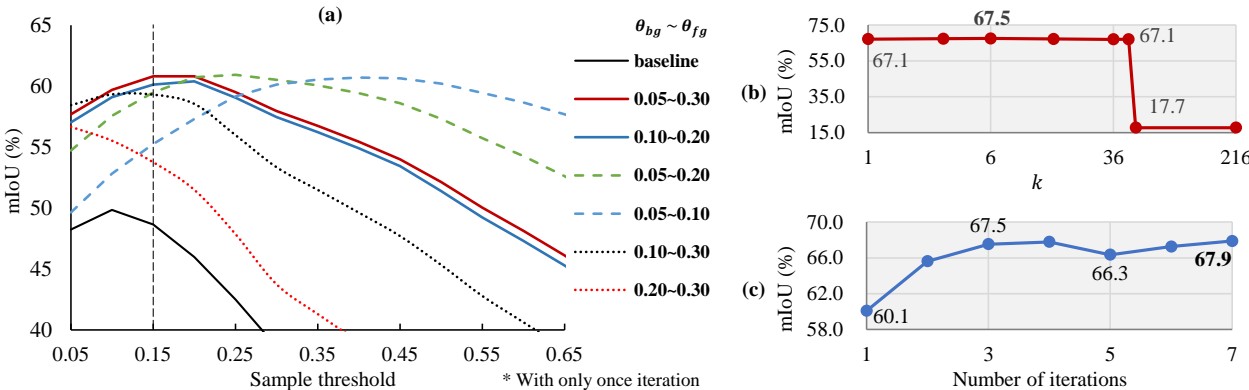

Figure 9: The CAM performance of GSLM with different (a) $\theta_{bg}, \theta_{fg}$, (b) $k$, and (c) number of iterations on PASCAL VOC 2012 *train* set.

to the baseline. In conclusion, the under-activation problem of CAM is reduced due to the reactivation of non-discriminating regions by the SLM of GSLM. Moreover, the over-activation problem is reduced by introducing boundary constraints in Coarse Generation.

**Effect of hyper-parameters.** Figure 9 reports the effects of some hyper-parameters on the CAM accuracy of GSLM, respectively.

- $\theta_{bg}, \theta_{fg}$: Figure 9(a) shows the CAM performance curve of GSLM for difference $\theta_{bg}, \theta_{fg}$. The sampling threshold is 0.15, with which baseline achieves 48.6% mIoU. When $(\theta_{bg}, \theta_{fg})$ is $(0.05, 0.3)$, the best threshold of GSLM coincides with the sampling threshold. However, decreasing $\theta_{fg}$ (dashed line item) or increasing $\theta_{bg}$ (dotted line item) makes the best threshold deviate from the sample threshold, which is not conducive to more iteration of GSLM. Narrowing the interval of $\theta_{bg}, \theta_{fg}$ (blue item) will slightly reduce performance.

- $k$: Figure 9(b) shows that GSLM is insensitive to $k$ when $k \leq 45$, and GSLM diverges when $k > 45$. This is due to the excessively large value of $k$, resulting in an excessively high output of the loss function Eq.(6).

- number of iterations: Figure 9(c) shows that the accuracy of GSLM is stable after 3 iterations.

- label compatibility of CRF: The label compatibility $\mu$ (using Potts model) of CRF (Krähenbühl & Koltun, 2011) indicates a penalty for nearby similar pixels that are classified as different class. A higher $\mu$ allows CRF to focus on refining class boundaries rather than correcting misclassifications, which complements the fine-grained CAM of GSLM after iterations. Therefore, we increase the $\mu$ from 3 to 10 in the last iteration of GSLM and GSLM+, resulting in a 0.6% mIoU improvement of masks for GSLM and 0.8% mIoU for GSLM+, respectively.

Table 6: The comparison of the foreground recall rate ($R_{fg}$), background recall rate ($R_{bg}$) and mIoU in % of CAM produced by DRS, SR (GSLM without iteration, i.e., line 5 of Table 5) and GSLM with (w/) and without (w/o) saliency maps ($\mathcal{S}$) on PASCAL VOC 2012 *train* set. All results of DRS are re-implemented.

| Method | $\mathbf{R_{fg}}$ | $\mathbf{R_{bg}}$ | mIoU |
|---|---|---|---|
| DRS w/o $\mathcal{S}$ | 70.1 | 78.3 | 42.2 |
| SR w/o $\mathcal{S}$ | 76.4 | 92.5 | 60.1 |
| GSLM w/o $\mathcal{S}$ | **84.9** | **92.8** | **67.5** |
| DRS w/ $\mathcal{S}$ | 72.5 | 96.0 | 66.3 |
| SR w/ $\mathcal{S}$ | 72.3 | **98.8** | 66.8 |
| GSLM w/ $\mathcal{S}$ | **81.1** | 97.8 | **72.4** |

**Comparison of Seed Reactivation and DRS.** In Section 4.3, we have introduced Seed Reactivation and contended it addresses the drawback of over-activation in DRS. In support of the statement, Table 6 reports the comparison of the CAM recall rates ($R = \frac{TP}{TP+FN}$, where $TP$ and $FN$ denotes true positive and false negative) of DRS, SR and GSLM. For fair comparison, we evaluate all methods with (w/) and without (w/o) saliency maps. The lower the background recall rate ($R_{bg}$), the more background regions mistakenly get high activation values, i.e., the more serious the over-activation. As shown in Table 6, without the background correction by saliency maps, the DRS only achieves a 78.33% on $R_{bg}$, resulting in a low mIoU of 42.2%. Saliency maps increase the $R_{bg}$ of DRS to 96.0%, yielding mIoU of 66.33% (this value is 62.9% reported by Kim et al. (2021)). SR obtains a high $R_{bg}$ of 92.5% without saliency maps, which proves the accurate activation of SR on the background. In addition, iteration can improve $R_{bg}$ of GSLM (from 76.4% to 84.9%) while maintaining the high $R_{fg}$ of SR, contributing to CAM precision.

# 6  Conclusion

In this paper, we aim to address WSSS by exploring and exploiting the Complementary Learning System (CLS) for CAM, which is mostly confined to small discriminating object areas. Specifically, we proposed General-Specific Learning Mechanism (GSLM) to help drive CAM advance in a fine-grained way. GSLM consists of General Learning Module (GLM) and Specific Learning Module (SLM). Specifically, GLM extracts localization information with boundary constraints from images and stores it as Confidence CAM. SLM organizes region reassigning guided by Confidence CAM, which reinforces salient regions and remarks confusion regions. Experimental results validate that our method achieves new state-of-art performance.

## Acknowledgement

This work was supported by the National Natural Science Foundation of China under Nos. 62276061. BH was supported by the NSFC General Program No. 62376235, Guangdong Basic and Applied Basic Research Foundation No. 2022A1515011652, HKBU Faculty Niche Research Areas No. RC-FNRA-IG/22-23/SCI/04, and HKBU CSD Departmental Incentive Scheme. TLL was partially supported by the following Australian Research Council projects: FT220100318, DP220102121, LP220100527, LP220200949, and IC190100031.

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
