# OpenReview forum: "Exploit CAM by itself: Complementary Learning System for Weakly Supervised Semantic Segmentation"
_TMLR — Accepted by TMLR_

### Review · Reviewer_tf98 · 2023-10-24

**Summary Of Contributions:**

This paper addresses the problem of generating fine-grained masks for semantic segmentation. It uses a Complementary Learning System and proposes a General-Specific Learning Mechanism. General-Specific Learning Mechanism contains a General Learning Module and a Specific Learning Module.  General Learning Module extracts localization representations from CAM.  The SLM expandsthe CAM in an explicit way.

**Audience:**

Yes

**Claims And Evidence:**

Yes

**Requested Changes:**

See weaknesses

**Strengths And Weaknesses:**

Strengths:
1. The paper is generally well-written, clearly structured, and quite easy to follow.
2. The target issues of the paper are meaningful and worth exploring. The motivation is clear.
3. This submission gives a valuable implementation of such an idea.

Weaknesses:
1. The results in Table 2 cannot demonstrate the advantage of this method. For example, GSLM+ achieves 70.6/70.6 in Val/Test, while L2G achieves 72.1/71.7.
2.  Using colorful images in Figure 8, and Figure 9 will be better.

---

### Review · Reviewer_oBpv · 2023-11-20

**Summary Of Contributions:**

The authors propose a method for weakly supervised sementic segmentation called GLSM. The approach relies on successive trainings of the classifier using a combination of image-level labels and pseudo-labelled segmentation masks.
The initial pseudo-labels are given by CAM, which is then used as ground truth in the second stage after some post-processing consisting of thresholding between very confident positives and negative pixels, and applying a CRF to infer the segmentation masks for uncertain pixels. The trainings are repeated until the CAM converges to the desired segmentation masks.
The authors evaluate their approach on Pascal VOC 2012 and MS-COCO 2014, and perform multiple ablation studies to show the impact of each of the components of their method.

**Audience:**

Yes

**Broader Impact Concerns:**

No broader impact concern, beyond the existing literature.

**Claims And Evidence:**

Yes

**Requested Changes:**

I believe this method is worthy of acceptance in TMLR as it seems both technically correct and will interest the audience. I particularly appreciated the ablation study. I invite the authors to list the state of the art more clearly, and relativize their claims for surpassing baselines in light of this.

Here are some additional questions that would benefit being answered in the paper:
* What are the hyperparameters used for the CRF and IRN?
* Figure 4. c) is confusing, as it represents the graph both before and after transformation, in different scales. I suggest to make the graph clearer by e.g. making two graphs for a clearer before / after visualization.
* Experimental setings: how are the image-level labels selected in VOC and COCO?

Minor:
page 2 missing parentheses "Seed Reactivation Eq. 4 is integrated" -> "Seed Reactivation (Eq. 4) is integrated"
page 5 "Artificial Intelligent" -> Artificial Intelligence

**Strengths And Weaknesses:**

Strengths:
* Simple method for refining the CAM towards segmentation masks which seem to yield good improvements with respect to CAM and competing methods.
* The method can be combined with other approaches and still lead to improvements (Table 4).
* The paper is well written and easy to follow for the most part.
* The use of many hyperparameters ($\theta_{fg}, \theta_{bg}, k, \alpha$) seem relatively tricky to adjust; however, their effect is well studied in the ablation study and answered most of my questions.

Weaknesses:
The paper is not exhaustive about the state-of-the art, and it seems multiple competing approaches using the same supervision reach significantly higher numbers than reported. For instance, "Boundary-enhanced Co-training for Weakly Supervised Semantic Segmentation" (Rong et al., CVPR 2023) reaches 73.7% mIoU on VOC-val, and multiple other examples can be found [1].
Unless I am mistaken about this, I invite the authors to be clearer about the WSSS state-of-the art while reporting their results.

[1] https://paperswithcode.com/sota/weakly-supervised-semantic-segmentation-on

---

### Review · Reviewer_hhvb · 2023-12-14

**Summary Of Contributions:**

This work aims to solve weakly supervised semantic segmentation(WSSS) with image-level labels by improving the generated Class Activation Maps(CAM). In particular, the authors propose two modules, General Learning Module, which produces the coarse CAMs and Specific Learning Module, which enhances the coarse CAMs with finer details. Overall, the method is competitive with prior state of the art approaches for WSSS.

**Audience:**

Yes

**Broader Impact Concerns:**

I do not see any broader impact concerns.

**Claims And Evidence:**

Yes

**Requested Changes:**

**Changes in writing**
1.  Improve the motivation of seed reactivation
2. Proper distinction with DRS should be made. Current distinction is not clear.
3. Improvement in activation loss description. Current one seems a bit confusing to me regarding its working.

**Experimental changes**
1.  Comparison with DRS as a baseline with multi stage training
2. Comparison with prior works in the main table

**Strengths And Weaknesses:**

Strength:
1. The method is simple and performs good on standard benchmarks.
2. The work has extensive results and ablations.

Weaknesses:
1. **Motivation of seed reactivation**
The author argue that the maximum pixel value maybe located in confusion regions of Confidence CAM (page 6 Section 4.3), which might result in failure in model convergence. I find it difficult to understand, how is this possible ? For conventional CAMs from equation 2, the highest value pixel for a given class map (c) , has to be in salient foreground as it is the highest pixel value. With max normalisation (per channel) as in eq 2, the activation value will be 1. How can this go in confusion region ? The motivation of seed reactivation feels confusing to me.

2.  **Similarity with DRS**
The seed reactivation (equation 4) which seems to be the key of the proposed approach is extremely similar to DRS. The authors try to delineate DRS and this work with the line - “Similarly, DRS (Kim et al., 2021) has been applied a relative upper bound to network feature maps to suppress the salient foreground with high confidence. However, the background may be incorrectly activated because the regions are not independent, as shown in Figure 4 (b).” The authors argue that backgrounds maybe incorrectly activated for DRS, which in my opinion should be explained more, as how it happens. Maybe a qualitative example should help (from real samples).

3. **Comparison with DRS**
The proposed approach comparison with DRS is missing. One baseline should also be comparison with DRS with training in different stages/iterations to show the impact of proposed seed reactivation as authors claim it to be superior to DRS.

4. **Comparison with prior works missing**
a. Weakly Supervised Semantic Segmentation by Pixel-to-Prototype Contrast, CVPR 22.
b. Out-of-Candidate Rectification for Weakly Supervised Semantic Segmentation, CVPR 23.
c. Boundary-enhanced Co-training for Weakly Supervised Semantic Segmentation, CVPR 2023.

5. **Understanding Activation loss**
The activation loss in equation 5, aims to bridge the GAP between improved CAM M’ and confidence CAM N. In this what is fixed and what has gradients (what is learnt ?) (M’ vs N). From what I understand, N is fixed which is given as supervision to improve M’ (which seems like to be the case from the figure). If that is true, then from the equation 5, non significant foreground regions in N with value 0, will pull the values in improved CAM M’ non significant foreground region towards 0, which seems to be conflicting towards the whole idea and also from Figure 4c. Can you provide more details on this.

---

> ### Author Response · Authors · 2023-12-30
> **Addressing Reviewer Feedback**
>
> **Q1:** Motivation of seed reactivation
>
> A1: Since the Confidence CAM ($N$) is computed from the CAM of the previous stage (via Equation 3), it is entirely possible that the confusion regions of $N$ contain the highest value pixel of this stage. There are some possible scenarios:
>
> - This stage does not use the same weights as the previous stage (even if the network structure of this stage may be different from the previous stage, for example, QLM of GSLM+ uses IRN, while SLM uses ResNet50), so the highest value pixel of this stage can appear anywhere.
>
> - Even if this stage uses the weight of the previous stage, it can only ensure that the highest value pixel is located in the salient foreground region of $N$ at the beginning of this stage training. With the progress of training, the salient foreground regions that this stage "considers" will change, for example, they may expand and occupy part of the confusion regions, which is exactly what we hope to see. In this case, the highest value pixel can be located in the confusion region of $N$.
>
> In summary, since the highest value pixel may be located in the confusion region and is ignored by activation loss (Equation 5), calculating CAM using Equation 2 does not achieve the reactivation of the confusion regions as expected.
>
> **Q2:** Similarity with DRS and qualitative examples for comparing DRS and SR.
>
> A2: Due to the limitation of space, the following comparative experiment between DRS and SR is not reported in the paper. As described in Section 4.3, DRS applies a relative upper bound to the feature maps, which truncates high activation values and thus suppresses the salient foreground, but at the same time, the activation values of all regions (including the background) are increased relatively. For example, a bound of 0.5 relatively increases all activation values by a factor of 2. These background regions with high activation values are more likely to be misclassified as foreground, resulting in overactivation. However, DRS neatly sidesteps this drawback by using extra supervision **saliency maps** as background masks. SR offers a constant upper bound $k$ and constrains the activation values of feature maps to $[0,k]$ through SLM training with activation loss. The change of each pixel in the feature map is independently related to $k$, so that the decrease of the activation values of the salient foreground regions will not lead to the increase of that of the background regions.
>
> In support of the above statement, the following table reports the CAM recall rates ($R=\frac{TP}{TP+FN}$) of DRS, SR (GSLM without iteration, i.e., line 5 of Table 5 in paper) and GSLM on PASCAL VOC 2012 train. To be fair, we compared recall rates for all methods with (w/) and without (w/o) saliency maps.
>
> | Method | $R_{fg}$ | $R_{bg}$ | mIoU |
> | ---- | ---- | ---- | ---- |
> | DRS w/o saliency maps | 70.12% | 78.33% | 42.24% |
> | SR w/o saliency maps | 76.43% | 92.46% | 60.08% |
> | GSLM w/o saliency maps | 84.92% | 92.76% | 67.50% |
> | - | - | - | - |
> | DRS w/ saliency maps | 72.48% | 95.97% | 66.33% |
> | SR w/ saliency maps | 72.28% | 98.79% | 66.78% |
> | GSLM w/ saliency maps | 81.08% | 97.80% | 72.39% |
>
> The lower the background recall rate ($R_{bg}$), the more background regions mistakenly get high activation values. Without the background correction by saliency maps, the DRS background recall rate was only 78.33%, resulting in a low accuracy of 42.24% mIoU. Saliency maps increased the $R_{bg}$ of DRS to 95.97%, yielding mIoU of 66.33% (this value was 62.9% in DRS paper). SR obtained a high background recall rate of 92.46% without saliency maps, which proves the accurate activation of SR on the background. In addition, iteration can improve $R_{fg}$ of GSLM (84.92%) while maintaining the high $R_{bg}$ of SR.
>
> We will include this experiment in the appendix of the updated version.
>
> **Q3:** Comparison with DRS.
>
> A3: Comparative experiments of DRS and SR are described in A2. We cannot iteratively train GSLM with DRS as a baseline because GSLM works with image-level labels only and does not provide the saliency maps required by DRS.
>
> **Q4:** Comparison with prior works missing.
>
> A4: We apologize for ignoring these excellent works and will include them in the updated version.
>
> **Q5:** Understanding Activation loss.
>
> A5: $N$ is fixed and is generated by the previous stage, which is given as supervision to improve $M’$. Non-significant foreground regions will not be pulled towards 0 because they are marked by $N$ as -1 (the "else" case of Equation 3) and thus ignored by activation loss (Equation 5) and do not participate in gradient descent.

---

### Decision · Action_Editor_y9K3 · 2024-01-29

**Recommendation:** Accept with minor revision

**Comment:**

The reviewers acknowledge that the method is simple and effective, and that the paper is clearly written. Nevertheless, the paper should be updated based on the authors' feedback to the initial reviews. The AE recommends the authors to:
- clarify the motivation for seed reactivation;
- incorporate the comparison to DRS;
- discuss and compare with the missing related works mentioned by reviewers oBpv and hhvb;
- clarify the activation loss in Eq. 5;
- discuss the hyper-parameter settings and sensitivity;
- clarify Figure 4;
- clarify the image-level experimental setting;
- fix the minor typos mentioned by reviewer oBpv;
- clarify the comparison with L2G;
- provide color versions of Figs. 8 and 9.

**Audience:**

The reviewers all acknowledge that the paper would be of interest to the community.

**Claims And Evidence:**

Most of the claims are supported by accurate, convincing, and clear evidence. The exception being that of "breakthrough improvements", which is debatable, as mentioned by the reviewers.

---

> ### Author Response · Authors · 2024-02-15
> **Updated based on the reviewers' comments**
>
> Thanks for comments of all reviewers and AE. We have submitted revisions based on the comments, details below:
>
> **1. Clarify the motivation for seed reactivation**
>
> Updated at Section 4.3 "Seed Reactivation" paragraph.
>
> **2. Incorporate the comparison to DRS**
>
> - Updated at  Section 4.3 "Seed Reactivation" paragraph.
>
> - Added Section 5.3 "Comparison of Seed Reactivation and DRS" paragraph
>
> **3. discuss and compare with the missing related works mentioned by reviewers oBpv and hhvb**
>
> - Updated Table3 and Table4 to include the missing realted works
>
> - Discussed and compared with the missing related works at Section 5.2 "Improvements on segmentation results" paragraph.
>
> **4. Clarify the activation loss in Eq. 5**
>
> Updated at Section 4.3 "Activation Loss" paragraph 2.
>
> **5. Discuss the hyper-parameter settings and sensitivity**
>
> Updated at the last item of 5.3 "Effect of hyper-parameters" paragraph
>
> **6. Clarify Figure 4**
>
> Updated Figure 4
>
> **7. clarify the image-level experimental setting**
>
> Updated at Section 5.1 "Dataset and Evaluation Metrics" paragraph line 7~8.
>
> **8. Fix the minor typos mentioned by reviewer oBpv**
>
> - Added page 2 missing parentheses.
>
> - Fixed typo "Artificial Intelligent" -> Artificial Intelligence.
>
> **9. Clarify the comparison with L2G**
>
> Updated at Section 5.2 "Improvements on segmentation results" paragraph line 7~9.
>
> **10. Provide color versions of Figs. 8 and 9**
>
> Updated Figure 8 and 9.

---

> > ### Comment · Action_Editor_y9K3 · 2024-02-29
> > **Additional revision**
> >
> > Dear Authors,
> >
> > Thank you for the revised version. Before the paper can be accepted, I would nonetheless ask you to tone down the claims of "breakthrough improvements", which were pointed out to be debatable be the reviewers.

---

> > > ### Author Response · Authors · 2024-02-29
> > > **We will update soon**
> > >
> > > Dear AE and reviewers,
> > >
> > > We are very sorry to about this claim, and will update soon.
> > >
> > > Senior author

---

> > > ### Author Response · Authors · 2024-02-29
> > > **Re: we have addressed the inappropriate description and updated**
> > >
> > > Dear AE and reviewers,
> > >
> > > We have addressed the concerns regarding the inappropriate description and submitted the updated version. Please let us know if any futher modifications are needed. Thanks for your assistence.
> > >
> > > Authors